# Trimodality Treatment of Superior Sulcus Non-Small Cell Lung Cancer: An Institutional Series of 47 Consecutive Patients

**Witold Rzyman** [1], **Małgorzata Łazar-Poniatowska** [2], **Robert Dziedzic** [1,*], **Tomasz Marjański** [1], **Mariusz Łapiński** [1] and **Rafał Dziadziuszko** [2]

1   Department of Thoracic Surgery, Faculty of Medicine, Medical University of Gdańsk, 80-210 Gdańsk, Poland
2   Department of Oncology and Radiotherapy, Faculty of Medicine, Medical University of Gdańsk, 80-210 Gdańsk, Poland
*   Correspondence: dziedzic@gumed.edu.pl; Tel.: +48-583493142

**Abstract:** Objectives: Treatment of superior sulcus tumors (SST) using concurrent chemoradiation followed by surgery is a current standard. However, due to the rarity of this entity, clinical experience in its treatment remains scarce. Here, we present the results of a large consecutive series of patients treated with concurrent chemoradiation followed by surgery at a single academic institution. Materials and Methods: The study group included 48 patients with pathologically confirmed SST. The treatment schedule consisted of preoperative 6-MV photon-beam radiotherapy (45–66 Gy delivered in 25–33 fractions over 5–6.5 weeks) and concurrent two cycles of platinum-based chemotherapy. Five weeks after completion of chemoradiation, pulmonary and chest wall resection was performed. Results: From 2006 to 2018, 47 of 48 consecutive patients meeting protocol criteria underwent two cycles of cisplatin-based chemotherapy and concurrent radiotherapy (45–66 Gy) followed by pulmonary resection. One patient did not undergo surgery due to brain metastases that occurred during induction therapy. The median follow-up was 64.7 months. Chemoradiation was well tolerated, with no toxicity-related deaths. Twenty-one patients (44%) developed grade 3–4 side effects, of which the most common was neutropenia (17 patients; 35.4%). Seventeen patients (36.2%) had postoperative complications, and 90-day mortality was 2.1%. Three- and five-year overall survival (OS) were 43.6% and 33.5%, respectively, and three- and five-year recurrence-free survival were 42.1% and 32.4%, respectively. Thirteen (27.7%) and 22 (46.8%) patients had a complete and major pathological response, respectively. Five-year OS in patients with complete tumor regression was 52.7% (95% CI 29.4–94.5). Predictive factors of long-term survival included age below 70 years, complete resection, pathological stage, and response to induction treatment. Conclusions: Chemoradiation followed by surgery is a relatively safe method with satisfactory outcomes.

**Keywords:** superior sulcus tumors; multimodality treatment; neoadjuvant chemoradiation; surgery; survival

## 1. Introduction

Superior sulcus tumor (SST), also known as Pancoast tumor, is a rare type of non-small cell lung cancer (NSCLC) that occurs in 2–5% of all lung cancer patients. The proximity of SST to vital thoracic inlet structures, including the brachial plexus, esophagus, spine, and supraclavicular vessels, makes its treatment particularly challenging in terms of oncological radicality and sufficient resection margins. Until the 1950s, SST was regarded as inoperable and was treated with 45–70 Gy radiotherapy alone, which resulted in a five-year survival (5YOS) of 5–10% [1]. In 1953, Chardak and McCallum reported that a patient treated with 65-Gy preoperative radiotherapy followed by surgery achieved progression-free survival (PFS) of five years [2]. In 1961, Shaw et al. reported a 24% overall survival (OS) in a series of 61 patients treated with 30 Gy/10 fractions radiotherapy followed by a resection [3]. Since then, this strategy has become the standard of care in

SST, with long-term OS varying between 25 and 30% [4]. In 2007, Rusch et al. published long-term results from the Southwest Oncology Group Trial 9416 (SWOG 9416) that used induction concurrent chemoradiation followed by surgery in SST [5]. Out of 110 patients who received two cycles of cisplatin and etoposide concurrently with 45-Gy radiotherapy, 76 were accepted for surgery. The complete pathological response was achieved in 56% of patients. The five-year OS in patients who received the full treatment protocol was 44% [5]. Following this publication, several studies reported real-world series with five years OS varying between 37% and 59% [6,7]. After these publications, potentially resectable superior sulcus tumors are the target of concurrent chemoradiotherapy followed by surgery. Such a strategy is recommended by European and American guidelines that are based mainly on a multicenter prospective phase II SWOG trial in North America, which demonstrated an excellent complete resection rate and markedly improved 5-year survival rates. Due to the fact that randomized controlled trials are difficult to perform in these rare tumors, other therapeutic options are considered in both guidelines [8,9].

At our institution, concurrent chemoradiotherapy followed by surgery was implemented as a routine SST treatment in 2007. We present the long-term results of a series of 47 patients treated with the trimodality treatment.

## 2. Materials and Methods

### 2.1. Patients

The data from consecutive patients with pathologically confirmed cT3/T4 N0/N1 superior sulcus NSCLC treated at the Medical University of Gdansk between 2007 and 2019 were retrospectively reviewed. SST was defined as cancer located at the top of the lung with the involvement of the apical chest wall above the level of the second rib [10]. Patients were staged according to the 8th Edition of the Union for International Cancer Control/American Joint Committee on Cancer TNM classification for NSCLC [11]. The pretreatment assessment included clinical evaluation, blood tests, bronchoscopy, chest computed tomography (CT) with upper abdomen evaluation, brain CT or magnetic resonance imaging (MRI), and pulmonary function tests as standard protocol for this type of tumor. Chest MRI has not been routinely performed. Since 2011, baseline 18-FDG positron emission tomography (PET)-CT has been mandatory. Each patient was discussed at a multidisciplinary tumor board meeting twice: before treatment and after preoperative therapy. All but one accepted patient completed the scheduled trimodality treatment of concurrent chemoradiation followed by surgery.

### 2.2. Treatment

#### 2.2.1. Preoperative Chemoradiation

All patients received 6-MV photon-beam radiotherapy with a linear accelerator, using the intensity-modulated radiation therapy technique. The median dose to the planning target volume prescribed according to the International Commission on Radiation Units and Measurements' guidelines was 45–66 Gy delivered in 25–33 fractions over 5–6.5 weeks. Concurrently with radiotherapy, patients were planned to receive two cycles of chemotherapy, which consisted of platinum doublets: cisplatin and etoposide, cisplatin and vinorelbine, carboplatin, and paclitaxel. Since the results of the PROCLAIM study were published [12], cisplatin and pemetrexed for non-squamous NSCLC were allowed. The choice of chemotherapy regimen was based mainly on patient comorbidities and the histopathological type of SST. Forty-seven out of 48 patients completed the trimodality treatment regimen and were assessed, as one patient developed brain metastases during preoperative therapy. This patient received further palliative treatment without surgery. Therefore, 47 patients, including 29 men and 18 women, with a mean age of 61 years (range, 42 to 74 years, Table 1) were analyzed for toxicity, response rates, surgical and pathologic results, PFS, and OS. Baseline patient characteristics are listed in Table 1.

**Table 1.** Baseline Patient Characteristics.

| Clinical Characteristics | Median (Range) or No. (%) |
|---|---|
| **Age (years)** | 61 (42–74) |
| **Sex** | |
| Male | 29 (61.7) |
| Female | 18 (38.3) |
| **Symptoms** | |
| Chest pain | 28 (59.6%) |
| Shoulder pain | 21 (44.7%) |
| Horner's syndrome | 4 (8.5%) |
| Asymptomatic | 6 (12.8%) |
| **Histology** | |
| Adenocarcinoma | 20 (42.6%) |
| Squamous cell carcinoma | 21 (44.7%) |
| Large-cell carcinoma | 3 (6.4%) |
| Adenosquamous carcinoma | 1 (2.1%) |
| Non otherwise specified non-small cell lung cancer | 2 (4.3%) |
| **Involved lung** | |
| Left | 15 |
| Right | 32 |
| **ypT stage VIIIth TNM** | |
| Tx | 7 (14.9%) |
| T1 | 1 (2.1%) |
| T2 | 0 |
| T3 | 31 (65.9%) |
| T4 | 8 (17.0%) |
| **ypN stage VIIIth TNM** | |
| N0 | 41 (87.2%) |
| N1 | 2 (4.2%) |
| N2 | 3 (6.4%) |
| N3 | 1 (2.1%) |

Data are presented as *n* (%). The total number of symptoms exceeds 47 due to the coexistence of symptoms.

### 2.2.2. Surgery

Forty-seven patients who received preoperative chemoradiotherapy were accepted for surgery. The surgical approach was chosen considering the extent, location, and local invasiveness of the primary tumor. According to Rami-Porta [13], resection was deemed complete when the following criteria were fulfilled: (1) free resection margins proved microscopically; (2) systematic nodal dissection or lobe-specific systematic nodal dissection; and (3) no extracapsular nodal extension of the tumor, and the highest mediastinal node removed to be negative. Vertebral body infiltration was a contraindication of surgery. In 41 patients, a lobectomy or lesser anatomic resection and *en bloc* chest wall resection, including one to six ribs, was performed.

Forty patients were operated on using the posterolateral Shaw-Paulson approach. In 4 patients, anterior incisions were performed in 2 Masaoka, and 2 hemi-clamshell incisions. In 3 patients, combined posterolateral incisions with the anterior Dartavelle approach were applied. One subclavian artery was resected and reconstructed by the graft, and in 3 patients, artery releasing from tumor involvement in the subadventitial plane was performed. In six patients with unsatisfactory performance status after induction chemoradiotherapy and with a complete or near-complete response, lobectomy without rib resection was performed, based on intraoperative evaluation of the possibility of such a procedure.

Before surgery, one prophylactic dose of 2 g cefazolin was routinely administered, followed by 7–10 days of piperacillin/tazobactam therapeutic dosing according to local bacteriological recommendations.

### *2.3. Response to Treatment*

Reassessment with chest CT was performed four weeks after completion of preoperative chemoradiation. Before surgery, patients underwent an evaluation that included a physical examination, performance status, blood tests, spirometry, a six-minute walk test, and a cardiovascular assessment. In the absence of general progression, patients were scheduled for surgery.

Patients undergoing pulmonary resection were restaged according to the ypTNM classification [11]. Pathological response in the primary lesion and regional lymph nodes was assessed using Junker's regression grading system: Grade I: no tumor regression or only spontaneous tumor regression; Grade IIa: morphologic evidence of therapy-induced tumor regression with at least 10% residual tumor cells presenting more than a focal microscopic disease; Grade IIb: less than 10% residual tumor cells presenting focal microscopic disease; Grade III: complete tumor regression with no evidence of vital tumor tissue in the sections of the primary lesion and mediastinal lymph nodes [14].

### *2.4. Follow-Up*

Patients were followed-up every three months for two years after surgery, and then every six to 12 months. Follow-up imaging included a CT every six months in the first two years, and annually thereafter. Two patients were lost to follow-up. The mortality rate was calculated based on data obtained from the National Lung Cancer Registry. Disease recurrence at the surgical margin, supraclavicular zone, ipsilateral hilum, and/or mediastinum was considered a local-regional failure. Relapse at other sites was considered a distant failure.

### *2.5. Statistical Analysis*

Study variables were summarized using descriptive statistics methods. Median with range and number with percentage were used to describe the characteristics of patients. The median follow-up time was calculated with a reverse Kaplan-Meier method. A Kaplan-Meier method with survival curves was used to estimate overall survival (OS) and recurrence-free survival (RFS). Log-rank test was used to compare the Kaplan-Meier estimates. RFS was calculated from the first day of radiotherapy to the date of confirmed progression or death from any cause. For patients who lacked information about the date of commencement of radiotherapy, the date of the surgery was assumed to be the commencement of treatment. Thirty and 90-day mortality rates and duration of postoperative stay were assessed. The univariate and multivariate Cox proportional hazard models were built to estimate the impact of the following parameters on OS: age $\geq$ 70, complete resection, pathological stage, and response to induction treatment. With a forward selection, the decision of whether to include or remove a covariate from the multivariate model at each iteration of the algorithm was based on univariable testing. A swimmer plot was used to show a subject's response to treatment, recurrence, and death over time. The statistical tests were two-sided, and a *p*-value < 0.05 was considered statistically significant. Calculations and statistical analyses were performed using R 4.0.4 statistical software (The R Foundation for Statistical Computing, Vienna, Austria).

## 3. Results

### *3.1. Preoperative Chemoradiation*

Thirty-eight patients received a total dose of 45 to 52 Gy in 25 to 26 fractions (Table 2). Nine patients with a risk of compromised surgical margins or ineligibility of surgery received a boost to a total dose of 60 to 66 Gy in 28 to 33 fractions. All patients received concurrently platinum-based chemotherapy (Table 2). Thirty-two patients received two planned chemotherapy cycles. In thirteen cases chemotherapy was ceased after the first cycle due to excessive toxicity, and in one case owing to patient refusal.

**Table 2.** Chemoradiotherapy—treatment.

| Chemotherapy Regimen *N* (%) | |
|---|---|
| cisplatin—etoposide | 41(87.2) |
| cisplatin—vinorelbine | 3 (6.4) |
| carboplatin (AUC 2)—paclitaxel weekly | 2 (4.2) |
| cisplatin—pemetrexed | 1 (2.1%) |
| **Number of chemotherapy cycles** | |
| 1 | 13 (27.7%) |
| 2 | 32 (68.1%) |
| >2 (5 cycles of carboplatin and paclitaxel) | 2 (4.2%) |
| **Radiotherapy dose** | |
| 45 Gy/25 fr. | 1 (2.1%) |
| 50 Gy/25 fr.–52 Gy/26 fr. | 37 (79.2%) |
| 60 Gy/30 fr.–66 Gy/30–33 fr. | 9 (18.8%) |

Data are presented as *n* (%).

Twenty-one patients (44%) developed grade 3–4 side effects following chemoradiation, of which the most common was neutropenia (17 patients; 35.4%). Sepsis, thrombocytopenia, pneumonia, and acute renal failure each occurred in one patient. Chemoradiotherapy was well tolerated with no toxicity-related deaths.

*3.2. Surgery*

Of the 47 operated patients, 41 underwent a lobectomy (87.2%); four bilobectomy (8.5%), and two segmentectomy (4.3%). *En bloc* chest wall resection was performed in 41 patients (87.2%). Six patients with complete or near complete radiological tumor regression on the CT image and poor performance status after induction chemoradiation underwent lobectomy without rib resection, after intraoperatively ascertaining the feasibility of this procedure. Infiltration in the apex was removed in the extrapleural plane (Table 3). In one patient, the subclavian artery was excised and reconstructed with a graft. In three cases with adherence of the tumor to the subclavian artery, subadventitial tumor detachment with branches ligation was performed. Complete mediastinal lymph node dissection was performed in all patients.

**Table 3.** Surgery—treatment.

| Pulmonary Resection | |
|---|---|
| segmentectomy I + II + III | 2 (4.2%) |
| upper lobectomy | 41 (87.2%) |
| bilobectomy | 4 (85%) |
| **Chest wall resection** | 41 (87.2%) |
| 1–2 ribs | 6 (12.8) |
| 3–6 ribs | 35 (74.5%) |
| extrapleural resection | 6 (12.8%) |
| vascular resection | 1 (2%) |
| **Pathologic response (Junker's grading)** | |
| grade IIa | 12 (25.5%) |
| grade IIb | 22 (46.8%) |
| grade III | 13 (27.7%) |

Data are presented as *n* (%). Junker's grading: Grade IIa: morphologic evidence of therapy-induced tumor regression with at least 10% residual tumor cells presenting more than focal microscopic disease; Grade IIb: less than 10% residual tumor cells presenting a focal microscopic disease; Grade III: complete tumor regression with no evidence of vital tumor tissue in the sections of the primary lesion and mediastinal lymph nodes.

In 42 of 47 patients (89.4%), resection was pathologically complete (R0). In 22 patients (46.8%), the grade IIB according to Junker's pathological response criteria with less than 10%

residual tumor in the postoperative specimen was observed. In 13 (27.7%) and 12 (25.5%) patients, grade III and IIA were assessed (Table 3).

Seventeen patients (36.2%) developed postoperative complications. The most frequent complications were arrhythmia in 10 (21.3%), and pneumonia in 5 (10.6%) patients, followed by atelectasis requiring frequent aspiration in 2 (5.2%). Respiratory insufficiency and myocardial infarction were diagnosed, respectively, in one patient each (2.1%). One patient (2.1%) died due to sudden multi-organ failure on postoperative day 11.

The median duration of the hospital stay at the surgery department was 13 days (range: 7 to 91 days).

### 3.3. Survival

Forty-seven patients were observed for a median of 64.7 months (95% CI 26.5–86.7) (Figure 1).

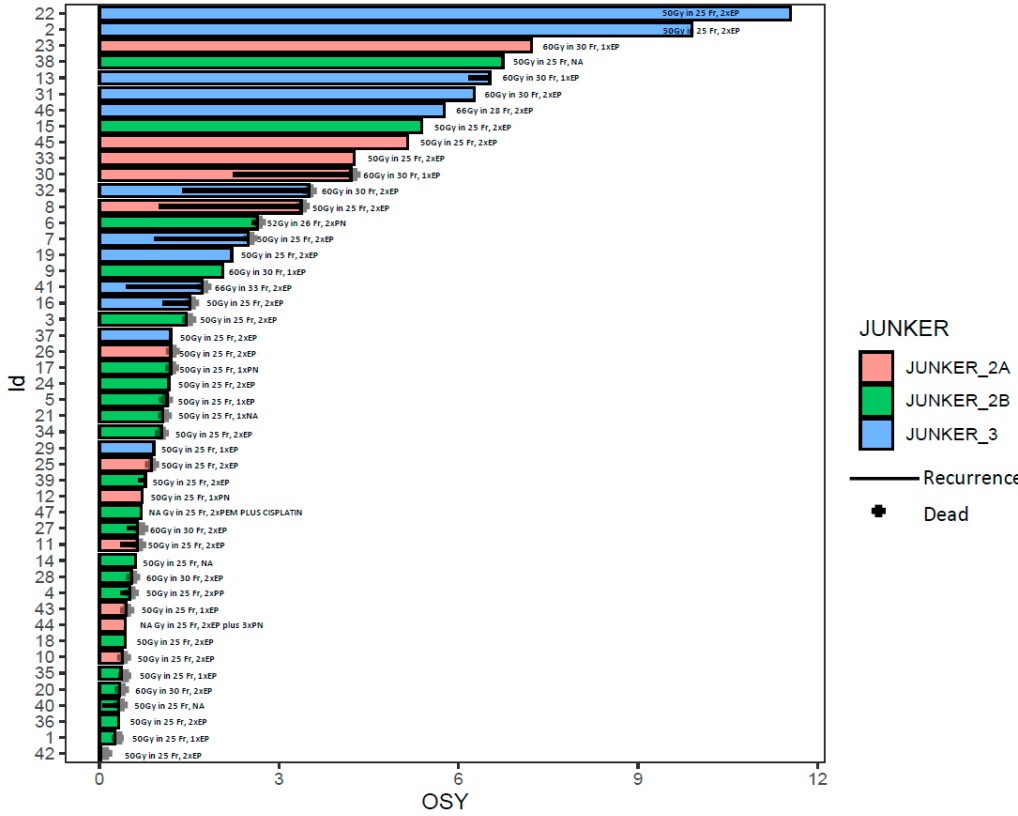

**Figure 1.** The swimmer plot shows patients' survival in years. Every patient is marked with a color according to pathological response to neoadjuvant treatment: Junker 2A—orange, Junker 2B—green, Junker 3—blue. The id is a unique number assigned to a patient. The black line is indicating the recurrence of the disease. The black cross—the death of a patient. OSY—overall survival in years.

Two patients were lost to follow-up. Thirty and 90-day mortality was 2.1%. Esti-mated one-, three- and five-year OS for the entire group was 72.9%, 43.6%, and 33.5%, respectively. Estimated one-, three- and five-year disease-free survival (DFS) was 65.8%, 42.1%, and 32.4%, respectively (Figure 2).

The three- and five-year OS of patients with R0 resection was 48.0% and 36.9%, respectively. In the univariate analysis, unfavorable prognostic factors for OS included stage ypT3 (HR: 3.5; 95% CI 1.4 -9.0), age above or equal to 70 years (HR: 4.3; 95% CI 1.6–11.8), incomplete resection (HR: 5.4; 95% CI 1.6–17.7), and Junker 2B (HR: 2.3; 95% CI 1.04–5.2). Complete pathological regression after chemoradiation (HR: 0.37; 95% CI 0.14–1.01) has not reached statistical significance in terms of longer OS.

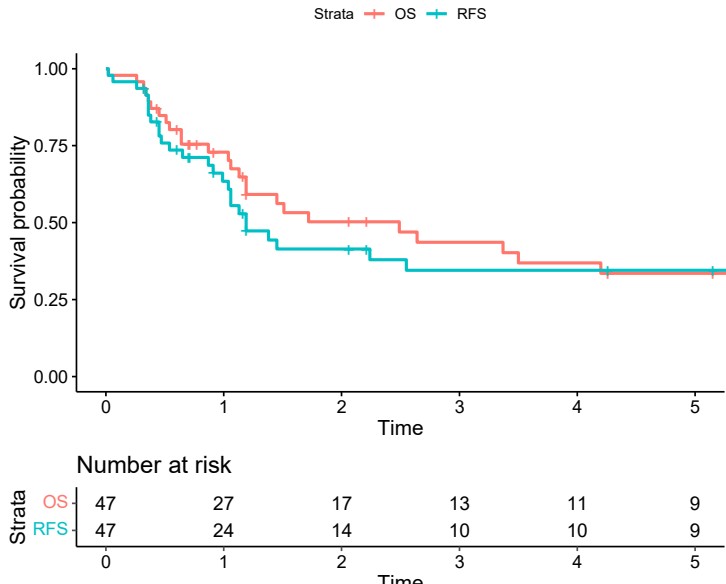

**Figure 2.** The Kaplan-Meier survival curves for overall survival (OS) are represented by a red line versus recurrence-free survival (RFS) represented by a blue line, with corresponding 95% confidence intervals illustrated with light-red and light-blue zones.

In the multivariate analysis, unfavorable prognostic factors for OS included the presence of local recurrence (HR: 3.4; 95% CI 1.6–7.5), Junkers 2B (HR: 3.2; 95% CI 1.2–8.9), incomplete resection (HR: 4.3; 95% CI 1.0–18.2), and age greater than 70 (HR: 3.5; 95% CI 1.1–11.3). Chest wall invasion after induction treatment—ypT3 (HR:2.7; 95% CI 0.9–7.4)—has not reached statistical significance (Figure 3).

| Variable | HR | 95% CI | p value | |
|---|---|---|---|---|
| Local recurrence | 3.4 | 1.6-7.5 | 0.002 | |
| Junkers 2B | 3.2 | 1.2-8.9 | 0.023 | |
| Positive resection margin -R1 | 4.3 | 1.0-18.2 | 0.045 | |
| ypT3 status | 2.7 | 0.9-7.4 | 0.066 | |
| Age grater than 70 | 3.5 | 1.1-11.3 | 0.035 | |

**Figure 3.** Multivariate analysis with forest plot of risk of death according to Cox proportional hazard ratio (HR).

The estimated 12-month OS of patients with R1 and R0 resection was 40% (95% CI 13.7–100) and 77.3% (95% CI 65.2–91.6), respectively (Figure 4). The five-year OS in the six patients without ribs resection was 20.8% (95% CI 36.8–100), compared to 36.1% (95% CI 22.2–58.6) for 41 with ribs resection. Distant recurrences dominated. The most common site of distant recurrence was the brain in six patients (12.8%) and the suprarenal glands in three (6.4%). Local recurrence occurred in six patients (6.4%), all of whom had R0 resection. The low number of patients with local recurrence does not allow for comparisons of other potential risk factors for relapse. Diagnosis of local recurrence was associated with a two times higher risk of death (Figure 4). For 13 patients with complete pathological response,

the three- and five-year OS was 63.3% (95% CI 39.9–100) and 52.7% (95% CI 29.4–94.5), respectively, and was significantly better than in partial or non-responders combined (34.3% (95% CI 19.8–59.4) and 24.5% (95% CI 11.9–50.5), respectively (Figures 3 and 4)). We have also calculated the OS in patients who were qualified for the trimodality treatment with and without PET examination. In 10 patients without preoperative PET, the OS was shorter (but not statistically significant) compared to the remaining group (HR 0.58, 95% CI 0.25–1.36, *p* = 0.21).

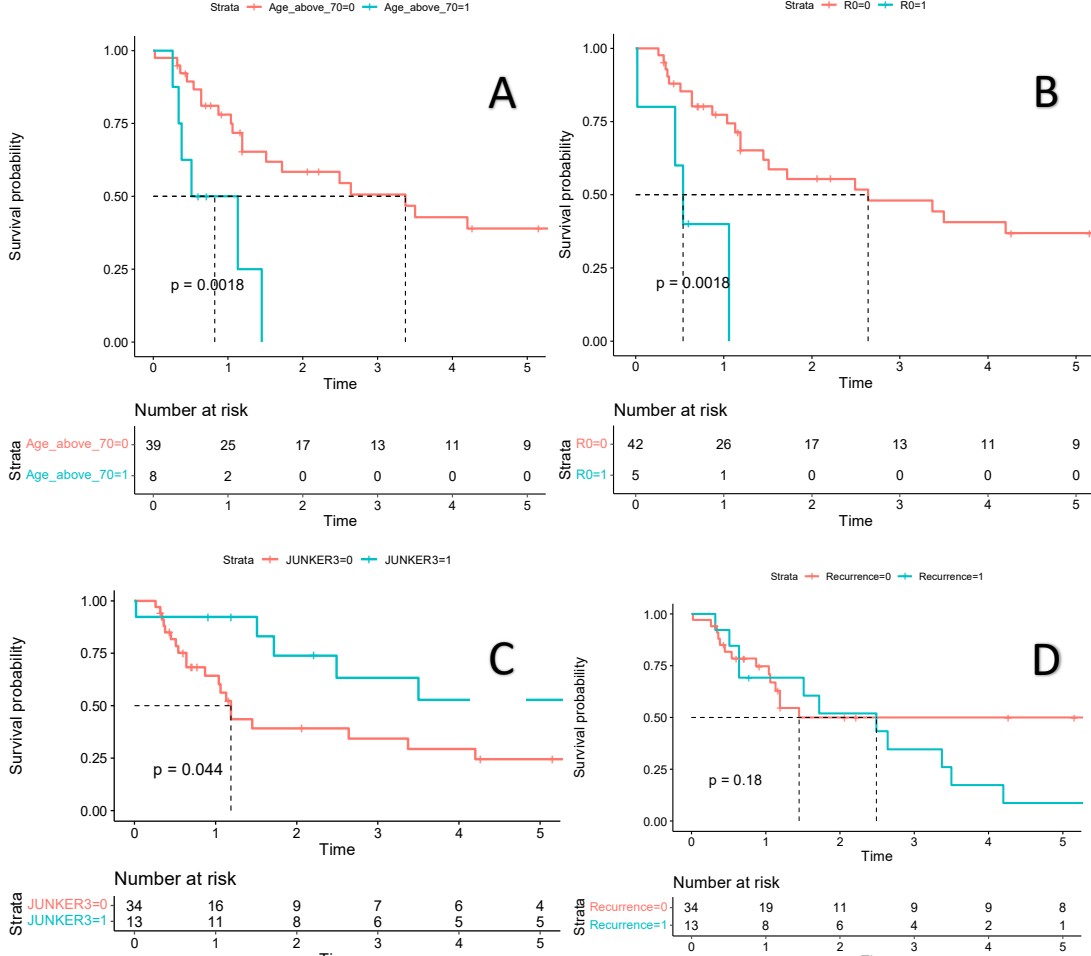

**Figure 4.** Estimated survival is presented with Kaplan-Meier survival curves. Plot (**A**)—Comparison of overall survival in a group of patients aged 70 years or older (blue line) versus a group of patients under 70 years of age (red line). Plot (**B**)—Comparison of overall survival in a group of patients with positive resection margin—R1 (blue line) versus a group of patients with negative resection margin—R0 (red line). Plot (**C**)—Comparison of overall survival in a group of patients with complete pathological regression according to JUNKER scale 3 (blue line) versus a group of patients without complete pathological regression—JUNKER 2a or 2b (red line). Plot (**D**)—Comparison of overall survival in a group of patients with recurrence of lung cancer (blue line) versus a group of patients without recurrence (red line). Figure legend: a red and blue line: estimated survival, light red and blue zone: 95% confidence intervals.

## 4. Discussion

Since the initial description of SST by Henry Pancoast in 1924, treatment has evolved significantly. Until the 1946 report by Walker on the successful irradiation of SST, these tumors were considered untreatable [1]. In 1961, preoperative radiotherapy followed by resection was performed in a series of 61 NSCLC patients, and this method became the

standard of care for subsequent decades [3]. Rusch et al. next revolutionized the standard of care in such clinical situations, with a report on the results of the SWOG 9416 study on the trimodality treatment of SST [5]. These results were significantly better than in previous reports using preoperative radiotherapy alone as an induction treatment. Subsequent reports of relatively small groups of patients treated using this strategy confirmed the effectiveness of the trimodality treatment. Yet, due to the relatively low incidences of these entities, only a few studies involve a larger number of patients. All these studies are listed in Table 4 [5–7,15–23].

**Table 4.** Major series of trimodality treatment of SST.

| Author | *N* | 5-Year OS | 5-Year DFS | Postoperative Complications | Perioperative Mortality |
|---|---|---|---|---|---|
| Rusch VW et al., 2007 [5] | 104 | 44% | NR | 52.3% | 1.8% |
| Pourel N et al., 2008 [15] | 72 | 40% (3-year) | NR | 43% | 6.9% (90-day) |
| Kunitoh et al., 2011 [6] | 57 | 56% | 45% | NR | 3.6% |
| Fischer S et al., 2008 [16] | 44 | 59% | NR | 45% | 5% |
| Kappers I et al., 2009 [17] | 22 | 37% | NR | NR | 0% |
| Kernstine KH et al., 2014 [7] | 29 | 61% (3-year) | 56% (3-year DFS) | 55% | 2.3% |
| Marra A et al., 2007 [18] | 29 | 46% | NR | 21% | 6.4% |
| Marulli G et al., 2015 [19] | 56 | 38% | NR | 10.7% | 5.4% |
| Wright CD et al., 2002 [23] | 15 | 84% (4-year) | NR | NR | 0% |
| Kwong KF et al., 2005 [20] | 36 | 50% | NR | NR | 2.7% |
| Collaud S et al., 2013 [21] | 48 | 61% | NR | 21% | 6% |
| Waseda R 2017., [22] | 46 | 63% | 45% | 19.6% | 0% |
| Present series | 47 | 34% | 32% | 36% | 2.1% (90-day) |

CRT—chemoradiation, NR—not reported, DFS—disease free survival, OS—overall survival.

This report summarizes our 13 years of experience with the trimodality treatment of SST. This is the first long-term retrospective series of consecutive patients treated at one institution. Our study showed that the trimodality treatment is a safe approach with an acceptable level of complication rate in each step, and low mortality.

In our study, the three-year OS of 43.6%, the five-year OS of 33.5%, and the five-year disease-free survival (DFS) of 32.4% are somewhat lower compared to the SWOG 9416 study and some other published series (Table 4) [5–7,15–23].

The operability rate of 97% in this study was exceptionally high. Hence, survival might have been affected because resection was attempted even in patients with reduced performance status after chemoradiation. However, the trimodality regimen in our series proved to be safe, considering 30- and 90-day mortality. One sudden, unexpected death was observed on the 11th postoperative day with symptoms of fulminating multiorgan failure that developed within 16 h. No direct cause of death was found in the autopsy.

As this study evaluated the real-world performance of the trimodality therapy, the poorer results can be explained by several reasons. First, the baseline PET CT examination was performed in only 37 patients (79%), before becoming a standard procedure in our diagnostic algorithm. Actually, there was a clear trend toward longer survival in patients who did undergo the baseline PET CT, compared to those who did not. Second, the entry criteria were less restrictive compared to those in prospective clinical trials. In consequence, patients might have presented with more locally advanced tumors and worse performance status.

In this series, patients were not restaged with PET CT before chest surgery. On the other hand, in NSCLC the role of PET CT for tumor restaging after preoperative treatment is controversial, and in most instances chest CT may be sufficiently accurate. However, SST is a high-risk entity and PET CT may be more sensitive in detecting early progression. Lastly, the relatively small number of patients in this series might have biased the results.

Complete pathological response after preoperative therapy, completeness of resection, and younger age are established favorable prognostic factors influencing long-term OS and DFS in SST [5–7]. Our results are concordant with these findings. Despite no significant difference in OS by the pathological response, the trend toward better survival in complete responders was apparent. The thoracic surgeon always attempts to excise the tumor with a wide margin of healthy tissue. However, this is not always feasible in SST due to invasion of the prevertebral lamina. Indeed, this was the main reason for incomplete resection in our patients. In such cases, detachment of the tumor in the space between the spine and the lamina results in positive surgical margins. An alternative solution is removing the vertebrae followed by spine stabilization. However, such management carries a considerable risk of additional complications and may be performed only in highly specialized centers. In most SST patients administered preoperative treatment, postoperative irradiation after incomplete resection is challenging due to earlier high-dose radiotherapy.

In our study, the treatment protocol included chemoradiation followed by surgery. Given that the downstaging of the tumor and the degree of pathological tumor regression are the most important prognostic factors, better outcomes may be expected by virtue of more effective preoperative treatment. For example, in the CheckMate 816 study, preoperative nivolumab plus chemotherapy resulted in a significantly longer event-free survival than chemotherapy alone (31.6 months vs. 20.8 months, respectively) ($p = 0.005$) [24]. Treatment outcomes may also be improved with novel adjuvant therapies, such as targeted therapies or immunotherapy. In a phase 3 ADAURA trial including *EGFR*- mutated stage IB to IIIA NSCLC, adjuvant third-generation EGFR tyrosine-kinase inhibitor osimertinib increased two-year disease-free survival to 89%, compared to 52% in the placebo group ($p < 0.001$), in patients with tumors showing *EGFR* mutation [25]. Similarly, in the IMpower 010 phase 3 trial, stage II-IIIA NSCLC patients whose tumors expressed programmed cell death-1 ligand (PD-L1), adjuvant atezolizumab (a monoclonal antibody targeting PD-L1) increased three-year disease-free survival rates to 60% compared to 48% with standard care ($p = 0.0039$) [26].

Our treatment protocol included radiochemotherapy followed by surgery. In most patients, genomic data were not available, at least not immediately after surgery. Since the majority of patients were diagnosed with squamous-cell carcinoma, and there were no registered targeted agents or immune checkpoint inhibitors available in this setting, genomic assessment was not routinely performed at initial diagnosis. Currently, more molecularly targeted therapies and immunotherapies are available, and the indications for molecular testing are broader. PD-L1 expression and next-generation sequencing are performed in virtually all NSCLC patients, using material taken before or after surgery. Future clinical trials in SST should focus on the optimal implementation of immune checkpoint inhibitors, either as a component of induction or consolidation treatment, or a combination thereof.

Our study confirms the significance of uniform protocols for patients undergoing the trimodality treatment of SST. There is also a need for detailed assessments of patient performance, and radiological and pathological response to preoperative therapy.

## 5. Conclusions

Despite recent developments, SST remains a therapeutic challenge. Induction chemoradiation followed by surgical resection remains the treatment of choice in this entity and provides better outcomes compared to other strategies. New therapeutic developments in NSCLC may increase treatment efficacy in this disease.

**Author Contributions:** Conceptualization, W.R. and R.D. (Rafał Dziadziuszko); methodology, W.R. and R.D. (Rafał Dziadziuszko); software, M.Ł.-P., M.Ł., T.M. and R.D. (Robert Dziedzic); validation M.Ł.-P., M.Ł., T.M. and R.D. (Robert Dziedzic); formal analysis, W.R., R.D. (Rafał Dziadziuszko) and R.D. (Robert Dziedzic); investigation, W.R., R.D. (Rafał Dziadziuszko), T.M. and R.D. (Robert Dziedzic); resources, M.Ł.-P., M.Ł., T.M. and R.D. (Robert Dziedzic); data curation, M.Ł.-P., M.Ł., T.M. and R.D. (Robert Dziedzic); writing—original draft preparation, W.R., R.D. (Rafał Dziadziuszko), M.Ł.-P., M.Ł., T.M. and R.D. (Robert Dziedzic); writing—review and editing, W.R., R.D. (Rafał Dziadziuszko), M.Ł.-P., M.Ł., T.M. and R.D. (Robert Dziedzic); visualization, R.D. (Robert Dziedzic); supervision, W.R. and R.D. (Rafał Dziadziuszko); project administration, M.Ł.-P.and M.Ł.; funding acquisition, M.Ł.-P. and M.Ł. All authors have read and agreed to the published version of the manuscript.

**Funding:** This research received no external funding.

**Institutional Review Board Statement:** The study was conducted in accordance with the Declaration of Helsinki, and approved by the Institutional Review Board of the Medical University of Gdansk (NKBBN/88/2016).

**Informed Consent Statement:** A retrospective analysis of prospectively gathered data and the lack of experimental intervention in the study group; the university institutional review board accepted the study and decided to waive informed consent (NKBBN/88/2016).

**Data Availability Statement:** Not applicable.

**Conflicts of Interest:** Rafał Dziadziuszko received honoraria from AstraZeneca, Roche, Novartis, Bristol Myers-Squibb, FoundationMedicine, Karyopharm, and Takeda and Boehringer Ingelheim. Tomasz Marjanski has received payment for consulting fees, lectures, and being an advisory board member of Roche Genentech. The cooperation is not related to the topic of this paper. The other authors have no conflict of interest to declare. The funders had no role in the design of the study; in the collection, analyses, or interpretation of data; in the writing of the manuscript; or in the decision to publish the results.

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
