# Peer review of "Trimodality Treatment of Superior Sulcus Non-Small Cell Lung Cancer: An Institutional Series of 47 Consecutive Patients"

_curroncol, doi:10.3390/curroncol30050344_

Round 1
Reviewer 1 Report
In this prospective study, Witold Rzyman et al. confirmed the trimodality treatment of superior sulcus non-small cell lung cancer in 47 consecutive patients. The authors documented the basic and treatment information of every patient in detail, again confirmed the treatment effect of this regimen.
However, this study cannot be published without the below listed modifications:
1. The information in Tables is difficult to read.
§ ‘Thirty-six patients received the total dose of 45 to 52 Gy in 25 to 26 fractions (Table 2). Eleven patients with a risk of compromised surgical margins or ineligibility of surgery received a boost to a total dose of 60 to 66 Gy in 28 to 33 fractions.’ Whereas the data in Table 2 shows 39 and 9, respectively.
§ In the ‘Chest wall resection’ section, are ‘extrapleural resection’, ‘Vascular resection’ also included in Chest wall resection? The authors should clarify that.
§ Pathologic response should be described before Complications. The description of Table 3 is too sketchy and ignores part of the important information, which makes the audience confused.
2. Starting from line 188, there is a lot of data being discussed but without data exhibition. Also, more background information is required for better understanding.
3. Table 4 included very significant clinical information, should be discussed more.
4. The structure of this entire study is chaotic and confusing, needs to be reorganized.
Author Response
Reviewer 1.
Replies to specific comments:
- The information in Tables is difficult to read.
- ‘Thirty-six patients received the total dose of 45 to 52 Gy in 25 to 26 fractions (Table 2). Eleven patients with a risk of compromised surgical margins or ineligibility of surgery received a boost to a total dose of 60 to 66 Gy in 28 to 33 fractions.’ Whereas the data in Table 2 shows 39 and 9, respectively
We appreciate this comment. We found additionally that the information in the text does not match information in tables, moreover actually there were some wrong information in one table regarding oncological treatment. This happened due to the fact that we have shown information of 48 patients instead of finally analyzed 47. One patient did not complete the entire course of treatment due to progression and the appearance of CNS metastases, so he should have been excluded from the analysis. We changed these figures. We have rearranged tab 1 and table 2.
- § In the ‘Chest wall resection’ section, are ‘extrapleural resection’, ‘Vascular resection’ also included in Chest wall resection? The authors should clarify that
We have removed “vascular resection” from the table. Vascular resection is performed with chest wall resection when the subclavian artery is infiltrated. It is described in the text. We agree that it was confusing when reading the table.
- Pathologic response should be described before Complications. The description of Table 3 is too sketchy and ignores part of the important information, which makes the audience confused.
The body of the text and tables content were reorganized as suggested by the reviewer. In the Material and Methods section pathological response is described before surgery. In the results section pathological response is shown only in the table. So, we discussed it and implemented it as a text before complications as suggested. We have added also interpretation of the data presented in table 3 in the “Results” section regarding induction chemoradiation
- Starting from line 188, there is a lot of data being discussed but without data exhibition. Also, more background information is required for better understanding.
The results section was reorganized to be clearer and background information was added.
- Table 4 included very significant clinical information, should be discussed more
We added and discussed the suggested topics more extensively in the discussion section
- The structure of this entire study is chaotic and confusing, needs to be reorganized.
The body of the text and tables configurations was changed extensively in order to try make the structure of the manuscript less confusing as suggested. We have moved some information from one table to another and changed the text. We have added two figures and rearranged tables with OS and RFS putting both in one figure.
Reviewer 2 Report
I have had the opportunity to review the article titled "Trimodality Treatment of Superior Sulcus non-Small Cell Lung Cancer: An Institutional Series of 47 Consecutive Patients" by Rzyman et al. Overall, the paper is novel and the language used is adequate. However, I recommend some edits in the statistical analysis to improve the quality of the paper.
The authors aimed to evaluate the outcomes of patients with superior sulcus non-small cell lung cancer treated with trimodality therapy. The study provides valuable information on the management of this subset of patients.
General comments :
- Introduction : The authors should include information the standard of care (if any) for these patients, based on US and European guidelines. It is great that they outline the results from previous trials.
- Methodology : The authors should provide more information in the statistical analysis section. The authors should provide more information on where and why and they have used Mann-Whitney U test. This non-parametric test can be used when comparing groups of numeric variables with skewed distribution and/or variance. The authors should mention whether these normality tests were performed and for which comparisons. I do not see any of these comparisons in the manuscript. Accordingly, the others should explain in which comparisons they used Fisher's exact or chi-2 test for categorical variables.
- Results : Figure 1A & 1B look identical to me. It is hard to believe that all the pts are deceased on the day of disease progression. The authors are advised to take a closer look at their data.
Suggestions :
- The authors, should provide the 1, 2 and 5 year DFS and RFS. Furthermore, the should perform KM plots and Cox-regression models for male vs female, smokers vs non-smokers, those having treatment toxicity vs not, and those that they achieved pCR or not. The same factors should be used in the multivariate analysis. Regarding the multivariate analysis, the authors should present the reference groups used to calculate the HR for the variables used. Furthermore, the should plot them used a forest plot to make the presentation of the manuscript less dry.
- The authors should also provide information regarding the response status of each pts on treatment based on the RESIST criteria. These categories should be used for KM plot analysis and to included in the multivariate analysis.
- Given the novelty of the paper, the authors should provide a swimmer plot for the pts cohort in order to describe the clinical course of each of the pts, including therapy info, imaging information, therapeutic response and relapse/death.
- The authors should include any available information regarding the genomic alterations of these pts. If there are information, it has to be used for KM plot analysis and to included in the multivariate analysis.
- Last but not least, due to the limited information available in the literature regarding this tumor, the authors should create a clinical nomogram for DFS and OS.
- Discussion : Are there any prognostic or predictive biomarkers for this disease or any molecular pathway involved in the pathogenesis? These points have to be discussed.
Author Response
Reviewer 2.
Replies to specific comments:
- Introduction: The authors should include information the standard of care (if any) for these patients, based on US and European guidelines. It is great that they outline the results from previous trials.
We have cited European and American guidelines in the introduction.
- Methodology: The authors should provide more information in the statistical analysis section. The authors should provide more information on where and why and they have used Mann-Whitney U test. This non-parametric test can be used when comparing groups of numeric variables with skewed distribution and/or variance. The authors should mention whether these normality tests were performed and for which comparisons. I do not see any of these comparisons in the manuscript. Accordingly, the others should explain in which comparisons they used Fisher's exact or chi-2 test for categorical variables.
We re-evaluated the statistical tools used in the analysis. The methodology chapter has been improved.
3. Results: Figure 1A & 1B look identical to me. It is hard to believe that all the pts is deceased on the day of disease progression. The authors are advised to take a closer look at their data.
We agree that the presented survival curves look similar. We reviewed available OS and RFS data for all patients included in the analysis. It is true that after a relapse, many patients die within a short time. We have also modified these two figures by presenting both OS and RFS in one figure for a better view and possible comparison (Table 2).
- The authors, should provide the 1, 2 and 5 year DFS and RFS. Furthermore, they should perform KM plots and Cox-regression models for male vs female, smokers vs non-smokers, those having treatment toxicity vs not, and those that they achieved pCR or not. The same factors should be used in the multivariate analysis. Regarding the multivariate analysis, the authors should present the reference groups used to calculate the HR for the variables used. Furthermore, the should plot them used a forest plot to make the presentation of the manuscript less dry.
We checked our data again, most of the analysis was re-calculated. We present the results as recommended by the reviewer in the form of tables in the revised manuscript.
- The authors should also provide information regarding the response status of each pts on treatment based on the RESIST criteria. These categories should be used for KM plot analysis and to include in the multivariate analysis.
We have removed the statement about RECIST assessment. Because of the surgical treatment that followed the first response assessment, response confirmation could not be done.
- Given the novelty of the paper, the authors should provide a swimmer plot for the pts cohort in order to describe the clinical course of each of the pts, including therapy info, imaging information, therapeutic response and relapse/death.
We have provided a swimmer plot concerning all treated patients. Figure 1.
- The authors should include any available information regarding the genomic alterations of these pts. If there are information, it has to be used for KM plot analysis and to include in the multivariate analysis.
In the vast majority of these patients’ genomic data was not available, at least not immediately after surgery. due to the lack of availability of adjuvant molecular targeted therapy was not available in Poland before 2018. Since the majority of patients were diagnosed with squamous-cell carcinomas, and there were no registered targeted agents or immune checkpoint inhibitors, genomic assessment and PD-L1 expression of the tumors was not routinely performed in this series at the time of initial diagnosis.”
- Last but not least, due to the limited information available in the literature regarding this tumor, the authors should create a clinical nomogram for DFS and OS.
Unfortunately, we were unable to create a clinical nomogram due to time constraints. This would require the creation of a new database and statistical re-analysis. Besides, with such a small number of patients, it is difficult to expect the creation of a valuable nomogram.
- Discussion: Are there any prognostic or predictive biomarkers for this disease or any molecular pathway involved in the pathogenesis? These points have to be discussed.
We have added suggested topics in the Discussion section
Round 2
Reviewer 1 Report
There are still issued remained unaddressed in this revised prospective study:
1. In Introduction, the authors listed too many examples to validate the feasibility of this Trimodality Treatment, which are not all necessary. Part of the examples can be deleted or combined into one or two sentences to simplify the reading.
2. The information in Tables is still hard to grasp:
2.1. The sequence of information being discussed in manuscript and in Tables should be consistent. In manuscript, the authors discussed Radiotherapy first then chemotherapy, whereas in Table 2 they did the opposite.
2.2. ‘In twelve cases chemotherapy was ceased after first cycle due to excessive toxicity and in one case owing to patient refusal.’ The number in Table 2 is 13.
2.3. In 3.1 and 3.2, the authors discussed the data jumping between Table 2 and Table 3. It is hard to follow. For example, ‘Twenty-one patients (44%) developed grade 3-4 side effects following chemoradiation, of which the most common was neutropenia (17 patients; 35.4% (Table 3))’. This information is discussed before Table 2.
2.4. The surgery part should be separated from Table 2 and form an independent Table.
2.5. ‘Infiltration in the apex was removed in the extrapleural plane’. Does this correspond to the ‘extrapleural resection’ in Table 2? No patient number is mentioned.
2.6. ‘In 42 of 47 patients (89.4%) resection was pathologically complete (R0)’, ‘A total of 35 patients (74,5%) achieved complete or near complete response after chemoradiation’, no information of these two was mentioned.
2.7. ‘Seventeen patients (36.2%) developed postoperative complications (Table 3). The most frequent complications were arrhythmia (20.8%) and pneumonia (10.4%). One patient (2.1%) died due to sudden multi-organ failure on postoperative day 11.’ Not consistent with Table 3 at all.
2.8. Pathological T and N status should be incorporated in patients’ characteristics.
3. 3.3 Results is hard to understand and follow.

Author Response
Open Review
( ) I would not like to sign my review report
(x) I would like to sign my review report
Quality of English Language
( ) English very difficult to understand/incomprehensible
( ) Extensive editing of English language and style required
(x) Moderate English changes required
( ) English language and style are fine/minor spell check required
( ) I am not qualified to assess the quality of English in this paper
Yes |
Can be improved |
Must be improved |
Not applicable |
|
Does the introduction provide sufficient background and include all relevant references? |
( ) |
( ) |
(x) |
( ) |
Are all the cited references relevant to the research? |
( ) |
( ) |
(x) |
( ) |
Is the research design appropriate? |
( ) |
( ) |
(x) |
( ) |
Are the methods adequately described? |
( ) |
( ) |
(x) |
( ) |
Are the results clearly presented? |
( ) |
( ) |
(x) |
( ) |
Are the conclusions supported by the results? |
( ) |
( ) |
(x) |
( ) |
Comments and Suggestions for Authors
There are still issued remained unaddressed in this revised prospective study:
- In Introduction, the authors listed too many examples to validate the feasibility of this Trimodality Treatment, which are not all necessary. Part of the examples can be deleted or combined into one or two sentences to simplify the reading.
We have modified the introduction
- The information in Tables is still hard to grasp:
2.1. The sequence of information being discussed in manuscript and in Tables should be consistent. In manuscript, the authors discussed Radiotherapy first then chemotherapy, whereas in Table 2 they did the opposite.
Information in the text was rearranged considering reviewer comment
2.2. ‘In twelve cases chemotherapy was ceased after first cycle due to excessive toxicity and in one case owing to patient refusal.’ The number in Table 2 is 13.
We have change it in the text
2.3. In 3.1 and 3.2, the authors discussed the data jumping between Table 2 and Table 3. It is hard to follow. For example, ‘Twenty-one patients (44%) developed grade 3-4 side effects following chemoradiation, of which the most common was neutropenia (17 patients; 35.4% (Table 3))’. This information is discussed before Table 2.
We have tided it up following reviewe’s suggestion
2.4. The surgery part should be separated from Table 2 and form an independent Table.
We have created a separate table for surgery (table 4)
2.5. ‘Infiltration in the apex was removed in the extrapleural plane’. Does this correspond to the ‘extrapleural resection’ in Table 2? No patient number is mentioned.
Yes, it does. There were 6 such patients. It is mentioned both in the text and in the table
2.6. ‘In 42 of 47 patients (89.4%) resection was pathologically complete (R0)’, ‘A total of 35 patients (74,5%) achieved complete or near complete response after chemoradiation’, no information of these two was mentioned.
These two are presented in the table separately but in the text both are summarized. Intention was to show how many patients have morphological response to preoperative treatment
2.7. ‘Seventeen patients (36.2%) developed postoperative complications (Table 3). The most frequent complications were arrhythmia (20.8%) and pneumonia (10.4%). One patient (2.1%) died due to sudden multi-organ failure on postoperative day 11.’ Not consistent with Table 3 at all.
Thank you. It was error in the text. We have corrected it.
2.8. Pathological T and N status should be incorporated in patients’ characteristics.
We have done as suggested.
- 3 Results is hard to understand and follow.
We have changed it to some extent. We do hope that now it looks better.
Reviewer 2 Report
The authors return a much improved version of their manuscript and have addressed all my comments. The methods are more clear and accurate, while the new analysis addresses critical comments missing in the old version, while being done accurately. I believe that the manuscript is ready for publication. Congratulation to the authors for their efforts.
Author Response
Thank you for positively accepting our responses to the review.
Round 3
Reviewer 1 Report
1. In general, the English still needs to be improved.
2. From line 194-197, the content was not exhibited in Table 3. The (Table 3) legend should be moved forward.
3.In Figure 1, there was no color showing. And the figure was partially cut off.
4. In Figure 2. there was no bright red or bright bright blue showing, the figure legend was very confusing.
Author Response
Thank you for your positive review.
- In general, the English still needs to be improved.
We reviewed the article again, language errors have been identified and corrected.
- From line 194-197, the content was not exhibited in Table 3. The (Table 3) legend should be moved forward.
We have moved the information regarding the Table 3.
- In Figure 1, there was no color showing. And the figure was partially cut off.
The Figure and the Figure 1 legend have been corrected.
- In Figure 2. there was no bright red or bright bright blue showing, the figure legend was very confusing.
The Figure 2 legend have been corrected.